# The Journey of a Medicinal Plant throughout Science: *Sphaeralcea angustifolia* (Cav.) G. Don (Malvaceae)

**DOI:** 10.3390/plants12020321

**Published:** 2023-01-10

**Authors:** Mariana Meckes-Fischer, Pilar Nicasio-Torres

**Affiliations:** 1Centro para el Diagnóstico en Metabolismo Energético y Medicina Mitocondrial (CEDIMEMM), Tlacoquemecatl 71, Int.2, Col. Del Valle, Benito Juárez, Ciudad de México 03100, Mexico; 2Centro de Investigación Biomédica del Sur (CIBIS), Instituto Mexicano del Seguro Social, Argentina No. 1 Col. Centro, Xochitepec 62790, Mexico

**Keywords:** *Sphaeralcea angustifolia*, coumarins, sphaeralcic acid, phytopharmaceutical, anti-arthritic, medicinal plants

## Abstract

Using herbal medicine is an ancestral cultural practice among Mexicans. A broad sector of society turns to plants to treat priority health problems, a reality that leads scientists to explore the healing value attributed to them. Advances in the experimental research of *Sphaeralcea angustifolia* confirmed the anti-inflammatory activity of the species; therefore, an analysis of the scope of these studies is now warranted. As such, this paper is a compendium of the advances published in the scientific literature (from 2004 to 2021) on the anti-inflammatory properties of this plant. The promise offered by the species as a potential therapeutic agent is also considered, without dismissing aspects necessary for the preservation of this resource and its cultural and physical environment. The chemical–pharmacological aspects of the wild plant and its in vitro culture are highlighted. The plant’s anti-inflammatory properties support its clinical application as an anti-inflammatory phytopharmaceutical to treat arthritic conditions. The sustained therapeutic potential of *S. angustifolia* is reinforced by the biotechnological processes designed to conserve the resource, thus contributing to the protection of biodiversity and cultural diversity, aspects distinctive of a megadiverse country such as Mexico.

## 1. Introduction

The historiography of medicinal plants in Mexico rescues the memory of indigenous peoples, their interaction with the dominant Western culture, and the transformation that pre-Hispanic experience underwent in the colonial period and modern times, which still shapes the Mexican territory and cultural dimension of Mexicans. The imprint of ancestral medical knowledge maintained by the 68 native ethnic groups that inhabit contemporary Mexico is based on a communion they establish with the plant world, the earth, and nature [1].

Extraordinary biological and cultural diversity are singularities that place Mexico among the megadiverse countries of our planet [2]. However, the exploitation and extinction of plant species that humans have brought upon ecological systems have consequently affected the cultural and physical environment of indigenous communities. This reality is appreciated today and significantly affects the persistence of knowledge on the therapeutic benefits of herbal resources. Therefore, when conceiving a research project on medicinal plants that aims to determine their therapeutic or economic potential, it is necessary to analyze the cultural background surrounding it and provide alternatives to protect a natural source of incalculable value.

Considering the ancestral trajectory that medicinal plants have in Mexico, and that this natural source is one of the remarkable remedies of traditional medicine (a medical system that, in Mexico, coexists with the official medical system), focusing the purpose of its study is a relevant factor and is necessary to keep in mind. To date, 4000–5000 medicinal species have been registered in Mexico; however, most have yet to be supported by scientific evidence [2].

## 2. Ethnobotanical Background of *S. angustifolia*

Based on ethnobotanical information from herbariums and scientific literature [3,4,5,6,7], a succinct botanical description of *S. angustifolia* was determined by Cavanilles and G. Don (Malvaceae). The herbaceous plant reaches up to 1.5 m high, grows erect with narrowly lanceolate leaves, and has racemose inflorescences with purple or pinkish-whitish petals, and the androecium is usually purple. The scientific name of the species is associated with the morphological characteristics it presents, etymologically derived from the Greek *sphaero* (globe) and *alcea* (mauve), the specific epithet *angustifolia* refers to its narrowly lanceolate leaves (Figure 1).

The common names adopted for plants in different regions and localities of the country are diverse. Around the “Mezquital” in Hidalgo Valley, the people named *S. angustifolia* “vara de San José”. In the State of Mexico, it is more frequently referred to as “hierba del Negro” and as “duraznillo cimarrón” in Querétaro. Other names include “malvón”, “cordón”, “hierba negra”, “hierba del golpe”, and “pintapan” [8,9,10]. In turn, the Nahuatl-speaking ethnic groups call it “tlixihuitl”, a literal translation of “Nigros’ herb” [11].

The species is native to the northern region of Mexico and has a wide distribution in the north and center of the Mexican territory, its wild propagation reaches to Colorado and Kansas in the south of the United States. Although it is a population that grows in spots as a natural control between plants, it is found in abundance in the Mexican states (Figure 2) of Aguascalientes, Chihuahua, Coahuila, Distrito Federal, Durango, Guanajuato, Hidalgo, Jalisco, Estado de México, Michoacán, Nuevo León, Puebla, Querétaro, San Luis Potosí, Sonora, Tamaulipas, Tlaxcala, Veracruz, and Zacatecas [3,4,6,7].

Concerning the healing properties attributed to *S. angustifolia*, and the popular use that the plant has in different areas of the country, it is frequently registered in ethnobotanical databases as an oral remedy to treat dysentery, stomach pain, and diarrhea. Likewise, references describe the use of the roots to treat diarrhea with blood, and baths prepared with the cooking of the branches are mentioned to combat stomach pain. In chronic diarrhea, a mixture of *S. angustifolia* with *Matricaria recutita* (L.) Fiori, *Buddleja scordioides* Kunth, and mint is also employed [9,10,11]. Although oral use of the plant to treat conditions that affect the gastrointestinal tract is frequent, the ethnobotanical information on the curative medicinal property attributed to “vara de San José” is mainly linked to the widespread topical use of the species to treat inflammatory processes and associated ailments. Thus, fresh leaves macerated in oil or fat are used in cases of bumps, cracks, and kinks. To reduce inflammation of angina, to combat rheumatism and bone pain, the leaves are fried in fat before covering the affected region with the preparation. The decoction of the plant or mixed with “arnica” (*Heterotheca inuloides* Cass) and “hierba del golpe” (*Lopezia racemosa* Cav.) is mentioned to wash wounds and strokes. In the states of Aguascalientes, Durango, and Guanajuato, its use is reported in cases of pain [9,10,11].

## 3. Experimental Studies on *S. angustifolia*

### 3.1. Selection, Collection, and First Trials

It is common for scientific researchers to choose a plant for their study based on the valuable ethnobotanical databases on Mexican medicinal plants created over several decades by Mexican academic institutions. Such was the case of *S. angustifolia* and its excursion through the laboratories and headquarters of the interdisciplinary team now in charge of validating the therapeutic potential of this plant.

Interest in *S. angustifolia* arose due to a pharmacological screening on the anti-inflammatory activity of some wild plant extracts selected according to the strategy described above. The study included an extract of “vara de San José”, a plant known for treating bruises and fractures proposed by oral information provided by Mr. Fidel Méndez. The specimens collected by researchers interested in the study of this plant were identified botanically by the ethnobotanist M.C. Abigaíl Aguilar Contreras, responsible for the Herbarium of Medicinal Plants of the Mexican Institute of Social Security at Mexico City (IMSSM). The chloroform extract of the aerial tissues of *S. angustifolia* effectively inhibited plantar edema formation induced with ƛ-carrageenan in the rat compared with controls using paired Student’s *t*-test (*p* = 0.05) [11]. These promising results prompted continued experimental research on one of the medicinal herbs that, until then, lacked chemical and pharmacological studies.

### 3.2. Chemical Assessment

Ontogeny plays an essential role in the nature of a plant’s secondary metabolites. In addition, the chemical composition in a specimen varies according to its habitat, geographical and climatological characteristics, and harvesting period. Most of the collections of *S. angustifolia* in the published studies were conducted in the State of Hidalgo; others came from the State of Mexico, Mexico City, and the State of San Luis Potosí. Analyzing the origin and date of the collections of *S. angustifolia* in the studies, it is possible to argue that these factors mentioned, such as the methodological nature, could explain the dissimilarity of the chemical composition that has been reported by some authors [12,13,14,15].

For the first time, in 2004, the chemical composition of a chloroform extract from aerial tissues (flowers, stems, and leaves) of the wild *S. angustifolia* plant grown in Caxuxi, Ixmiquilpan Village (State of Hidalgo) was reported [12]. A chemical separation led to the isolation of the hydroxycoumarins scopoletin, esculetin and esculin, transcinnamic acid, the mixture of α- and β-amyrine, stigmasterol, and β-sitosterol (Table 1). Scopoletin was the most abundant compound in the extract. A different chemical composition (flavonoids apigenin and tiliroside, protocatechuic acid, caffeic acid, and β-sitosterol) was later reported for the ethanolic extract of the species collected in Tláhuac, México [13]. In addition, the species collected in the municipalities of Epazoyucan and Pachuca at the State of Hidalgo reported the lactone (-)-loliolide [14] in the methanol extract, and the compounds β-eudesmol and phytol were identified in the hexane extract of the flowers [15].

Significant advances in chemical studies of *S. angustifolia* occurred when biotechnological procedures were applied to obtain in vitro cultures of cells in suspension from plant leaves cultivated in the appropriate media (Figure 3). Cells grown in suspension preserved the capacity to produce scopoletin. Two other compounds (Table 2) that had not been previously identified in the wild plant were isolated from the dichloromethane: methanol extract, tomentin, and sphaeralcic acid (naftoic [2-(1,8-dihydroxy-4-isopropyl-6-methyl-7-methoxy) acid) [16].

**Table 1 plants-12-00321-t001:** Compounds identified in the extracts of aerial tissues of *Sphaeralcea angustifolia* wild plant.

Extract	Biological Effect	Compounds	References
Dichloromethane	Anti-inflammatoryImmunomodulatory	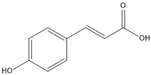	*p*-Cumaric acid	[12,17,18]
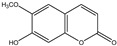	Scopoletin
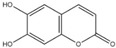	Esculetin
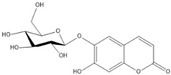	Esculin
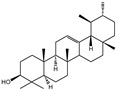	α-Amyrin
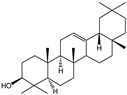	β-Amyrin
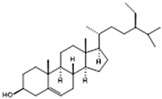	β-Sitosterol
Ethanolic	Antiprotozoal	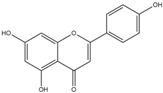	Apigenin	[13]
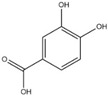	Protocatechuic acid
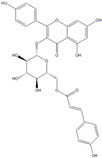	Tiliroside
		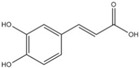	Caffeic acid
Methanolic	Neuroprotective	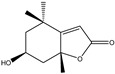	Loliolide	[14,15]
Hexanic	Herbicid	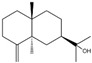	*Β*-Eudesmol
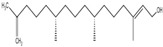	Phytol

### 3.3. Anti-Inflammatory Evaluation

The pharmacological study of the extracts and compounds mentioned above was conducted to investigate the anti-inflammatory properties popularly attributed to the plant. For this purpose, classic models that reported on acute and chronic inflammation in different strains of rats and mice were implemented. The animal models remain a valuable tool for discovering new mechanisms and therapeutic principles [25,26,27].

Reports on *S. angustifolia* report on the use of an arthritis model induced with complete Freund’s adjuvant (CFA) with an attenuated strain of *Mycobacterium tuberculosis* by subplantar administration. The dichloromethane extract from aerial tissues of the wild plant was administered ip at 100 mg/kg for 8 days and produced a sustained and significant inhibition of the edema (62.6%). Topical application of the dichloromethane extract also reduced ear edema (50.6%) and exhibited a protective effect against 12-O-tetradecanoyl phorbol-13-acetate (TPA)-induced mouse ear irritation. Both effects obtained with *S. angustifolia* extract were compared with the Student’s *t*-test with a *p* < 0.05 with respect to those mice of the control group. In turn, the assessment of the products derived from the chemical fractionation of the extract confirmed scopoletin as the most abundant component in the fraction with the greatest anti-inflammatory activity [12].

The activity of the dichloromethane extract (100 mg/kg, i.p.) of the wild plant was reproduced in the rat model of arthritis induced with CFA emulsion: PBS (phosphate buffer saline) by intradermal application at the base of the tail. After measuring the inflammation of the knee joint produced by the CFA, blood samples were obtained from the animals’ portal vein and synovial fluid. The ELISA technique demonstrated an inhibitory effect of the extract on the biosynthesis of the pro-inflammatory interleukins IL-1β, IL-6, and TNF-α while increasing the levels of the anti-inflammatory IL-10. Each IL level of mice treated with *S. angustifolia* extract was compared with those of mice of the control group, using a Student’s *t*-test with a *p* < 0.05 [17]. Furthermore, based on microarray results, the extract regulates the inflammatory process through the down-regulation of the expression of NF-ƙB, IL-12, IL-12R, CXCR4, IL-6, MMP9, MMP13, Pik3r2, and Pik3cb and the over-expression of IL-1Rα (IL-1 antagonist) and Hsp70, which stimulates IL-10 production [18].

The activity of products derived from cells in a suspension culture was assessed in models of acute inflammation induced with ƛ-carrageenan in the mouse plantar area and with TPA in the mouse ear. Significant differences were found among the treatment inhibition percentages by ANOVA and Tukey test [16]. The results showed that the administration of 100 mg/kg (i.p.) of the dichloromethane:methanol extract (9:1) from the biomass of cells in suspension inhibited the induced plantar edema, with activity lower than that recorded with 400 mg/kg of the wild plant extract. The extract of the culture medium and cells in suspension grown in Murashige and Skoog (MS) medium with nitrate restriction (2.74 mmol) presented a similar anti-edematous activity in the mouse plantar model. The biomass extract showed a dose-dependent effect with a median effective dose (ED_50_) = 137.63 mg/kg. Tomentin and sphaeralcic acid (45 mg/kg; i.p.) showed a similar inhibition on the plantar and auricular edema to that of indomethacin (0.5 mg), and sphaeralcic acid showed a dose-dependent effect at ED_50_ = 0.54 mg in TPA ear edema [16].

The activity of the biomass extract (100 mg/kg, standardized in scopoletin, tomentin and sphaeralcic acid), tomentin (93% purity), and sphaeralcic acid (98% purity) with both compounds at 5.0, 10.0, 15.0, and 20.0 mg/kg, were also tested in an experimental monoarthritis model induced by kaolin/ƛ-carrageenan. The anti-inflammatory effect produced by daily oral administration (9 days) was determined based on the size of the joint edema produced, and the expression of the pro-inflammatory cytokines IL-1β and TNF-α, and the anti-inflammatory IL-10 and IL-4 measured by the ELISA technique in the joint tissue. The extract produced a sustained decrease in the edematous process, with the affected extremities showing a normal appearance on the ninth day of treatment. The anti-edematous effect of the extract was 72%, and the activity of tomentin (ED_50_ = 10.32 mg/kg) and sphaeralcic acid (ED_50_ = 7.8 mg/kg) was dose-dependent and similar to that of methotrexate. Significant differences were found among the treatment inhibition percentages by ANOVA and Dunnett’s test. In turn, the expression of the interleukins IL-1β and TNF-α decreased significantly (Student´s *t*-test *p* < 0.05) in the joint tissue of the groups treated with the extract, sphaeralcic acid, and tomentin in comparison to the control group; being significantly higher than the expression presented by the anti-inflammatory interleukin IL-10. A probable immunomodulatory effect was attributed to the compounds [19,20].

### 3.4. Pharmacokinetic Profile

Pharmacokinetics is a pharmacological tool that evaluates the processes of absorption, distribution, metabolism, and elimination of the active products of a drug or a phytopharmaceutical that can eventually be transformed after oral or intravenous administration. The pharmacokinetics of a fraction of the cell suspension culture of *S. angustifolia* enriched in a mixture of scopoletin, tomentin (7 mg) and sphaeralcic acid (35 mg) were established by HPLC analysis. The detection and time course of the compounds were followed in the plasma, urine, and feces of mice after a single oral dose (400 mg/kg) of the standardized fraction. The authors describe the extraction of the biomass and its chemical fractionation to obtain the standardized active fraction, HPLC analysis, and the extraction and stability of the active compounds based on the guidelines established by the FDA (Food and Drug Administration) for plasma samples [28]. The pharmacokinetic profile of the mixture of coumarins and sphaeralcic acid showed that these compounds were detected in mouse plasma, and no products of their biotransformation were registered. It was determined that coumarins and sphaeralcic acid circulate to the white site and are subsequently detected in urine and feces. It was concluded that the mixture of scopoletin and tomentin are eliminated by the renal route and sphaeralcic acid by the enterohepatic route [28,29].

### 3.5. Clinical Trials of S. angustifolia Wild Plant Extract

Preclinical advances using standardized plant extracts were a starting point for two pilot clinical studies in hand and knee osteoarthritis patients. The report on patients affected by osteoarthritis of the hand [30] presents the results of a randomized, double-blind trial of a gel formulation made with 1% of a dichloromethane extract from *S. angustifolia* aerial tissues standardized in scopoletin. The group treated with the phytodrug showed therapeutic effectiveness and tolerability statistically similar to that of the control group of patients treated with 2% diclofenac. In both groups, the pain and inflammation decreased, and stiffness of the joints was observed, as well as the restoration of movement and closure of the hand. The extract produced no adverse effects. The authors attribute the effect to scopoletin, whose anti-inflammatory and anti-arthritic activity has already been reported [21,22,23], without ruling out the presence of other bioactive compounds in the extract. In Mexico, knee osteoarthritis is a common type of arthropathy that affects a large majority of older adults. Clinically, the condition is associated with pain, inflammation, stiffness, deformity and loss of functionality, symptoms that require treatment for relief. This can be caused by inflammation of the soft tissues around the joint.

A study of the effectiveness of *S. angustifolia* extract in patients diagnosed with osteoarthritis and knee pain is currently underway.

### 3.6. Biotechnological Procedures

As stated by the National Commission for the Knowledge and Use of Biodiversity (CONABIO), the degradation and loss of ecosystems, the overexploitation of species for consumption or commercialization, the introduction of exotic and invasive plants, pollution, climate change, and urbanization processes are risk factors that affect biological diversity [31,32]. Since *S. angustifolia* grows as isolated patches, especially from April to June, biotechnological techniques were implemented from the beginning of the project. In vitro cultures were established to obtain cultures of cells in suspension to find metabolites of clinical importance and for the micropropagation of the plant to ensure its survival (Figure 3).

To increase the content of the anti-inflammatory compounds detected in the aerial parts of the wild plant, in vitro conditions were established to obtain dedifferentiated tissue (callus) from leaf explants in MS medium with naphthalenacetic acid (NAA, 0.5, 1, and 2 mg/L) in combination with kinetin (Kin, 0.1 mg/L). The shoots generated in the callus (0.5 mg/L of NAA) were elongated in MS medium without growth regulators and acclimated to greenhouse plant growth conditions. The absence of scopoletin in the callus led to the development of cell suspension cultures using liquid MS medium supplemented with 2 mg/L of NAA and 0.1 mg/L of Kin. The cells in suspension accumulated scopoletin which was also released into the culture medium (0.11 mg/L) in low concentrations [16,33].

As a phenolic compound, scopoletin has a carbon-rich structure; therefore, secondary metabolism in the cellular suspension of *S. angustifolia* was promoted in MS medium by restricting the nitrate concentration from 27.4 mM to 2.74 mM. Under these conditions, the production of scopoletin (0.038%) increased 58-fold relative to the aerial tissues of the wild plant (0.00067%). Furthermore, 5-hydroxy-6,7-dimetoxicumarine known as tomentin (0.003%) and sphaeralcic acid, 2-(1,8-dihydroxy-4-isopropyl-6-methyl-7-methoxy) naftoic acid (0.004%) compounds were produced and for the first time reported in the genus *Sphaeralcea*. The methodology followed for developing the cellular suspension from friable calluses of *S. angustifolia* leaf explants is described in detail in [16]. This study gave rise to a patent.

It is well known that coumarins are phytoalexins that accumulate in plants under stressful conditions to stimulate the production of defense compounds. To increase the production of the active metabolites of *S. angustifolia* in the culture of cells in suspension, abiotic stress was produced with copper. The effect that the nitrate and copper content in the medium exerts on the growth of cells in suspension and the production of active compounds was analyzed by applying the statistical design (a 2^K^ factorial design and a central composite design) of the response surface methodology. In short, it was determined that the interaction of 2.74 mM of nitrate and 2 µM of copper in the culture medium at 2–4 days produced the highest content of the mixture of coumarins (4.14 mg/L) and sphaeralcic acid (1.44 mg/L). However, according to the CCD (Central Composite Design) test, at a concentration of 2.42 mM of nitrate and 2.35 µM of copper, the maximum content of sphaeralcic acid would be increased to 6.1 mg/L without affecting cell growth, a situation that would allow large-scale cultivation of suspended cells in a stirred-tank bioreactor [34].

Given that the compounds are accumulated in cells and are also excreted into the culture medium, the conditions of the culture of the cells in suspension in MS medium with 2.74 mM of total nitrate were explored in 2 L stirred tanks (Applikon, Schiedam, Netherlands) operated with different stirring speeds and aeration volume. The maximum growth of the suspended cells (19.11 g/L) was recorded 11 days after culture at 200 rpm. Sphaeralcic acid was the primary intracellular compound whose content (52.15 mg/L) was significantly higher than that reported in cells developed in Erlenmeyer flasks [35].

The genetically transformed root cultures are considered an important biotechnological tool for active compound production because they are genetic and biochemical stables, attributes that give them advantages over cell suspension cultures. Transformed root cultures of *S. angustifolia* mediated by *Agrobacterium rhizogenes* ATCC15834/pTDT were generated from nodal segments of plantlets (Figure 3). Five lines of the hairy roots had the potential to produce scopoletin and sphaeralcic acid, both accumulated in the cells and excreted to the culture medium. The anti-inflammatory compound production was achieved over 2 years; subsequently, this biotechnological system was proposed to scale up, projecting its commercial production [24]. The SaTR N7.2 line presented the highest sphaeralcic acid production (17.60 ± 1.72 mg/g); this production was 440-fold superior to that reported in *S. angustifolia* wild plants, and it was 263-fold higher than the cells in the suspension of *S. angustifolia* cultivated in MS medium with nitrate restriction in flasks, and 5-fold higher when it was cultivated in a stirred-tank type bioreactor.

## 4. Discussion

The bibliographic collections on medicinal plants used in Mexico in the past show the use of *S. angustifolia* to treat inflammatory processes such as tonsillitis, bronchitis, conjunctivitis, gastritis, enteritis, contusions, and hemorrhoids [36]. In the same way, the ethnobotanical fieldwork conducted from the last decades of the 20th century allow medicinal herbalism in ethnobotanical data banks to be updated, and confirms the repeated and continuous use of this plant as an anti-inflammatory agent.

Laboratory animal models are especially useful for evaluating therapeutics in the preclinical phase. If acceptable results clarify the pharmacological effect of the product, a clinical trial is one step closer. The activity of *S. angustifolia* assessed at a laboratory stage in the models described by the authors confirmed the properties popularly attributed to the plant. Furthermore, the studies that applied the standardized plant-based phytopharmaceuticals in patients affected by hand osteoarthritis established for the first time the safety, efficacy, and tolerability of the drug. Furthermore, the traditional topical use of the plant reproduces the effect.

On the other hand, the coumarins (scopoletin, tomentin) and naphthoic acid identified in wild plants and in vitro cultures were considered to be the active compounds that support pharmacological and clinical studies; therefore, the formulations are standardized. Clinical studies show that phytopharmaceuticals reproduce the anti-osteoarthritic activity of some drugs frequently used to treat patients’ ailments.

Since the beginning of current pharmaceuticals, natural products have played a crucial role in the search for drugs required by the industry. However, for pharmaceutical companies, the attractiveness of drug development from natural compounds weakened at the end of the twentieth century [36,37,38,39]. The introduction of highly sophisticated technological procedures, such as high-throughput screening (HTS), and combinatorial chemistry, provided alternatives that significantly augured the discovery of new molecules with therapeutic benefit [1,40,41,42].

Between 1981 and 2014, one-third of the new drugs on the market were derived from natural sources, 25% of which came from plants [38]. Furthermore, the chemical structure of many of the synthetic drugs obtained reproduced that of compounds of natural origin. Although few new compounds have been approved for clinical application, in the search for less toxic and highly specific drugs, innovative technologies will continue to play a relevant role and maintain the expectations and future projections of large pharmaceutical companies [2].

Ethnopharmacological research on medicinal herbs in Mexico goes beyond the interest in explaining, based on scientific support, the healing properties of the plants that the population uses to address health problems. Thus, the knowledge generated in laboratory tests around this important natural source is also a common thread for discovering new therapeutic agents, either molecules or phytomedicines standardized in the active components. For many reasons, among them cultural, phytopharmaceuticals are already greatly accepted by the population and, step-by-step, have also been gaining traction among clinicians who do not reject the benefits provided by these drugs as additions to therapy [43].

The loss of medicinal plants is a worldwide phenomenon widely reported and attributed to urbanization, deforestation, and species overexploitation for drug discovery and the production of remedies. In Mexico, it is important to highlight that valuable medicinal species grow wild, so many of them are potentially at risk of being threatened by their indiscriminate harvesting [44]. Culturing medicinal plants under experimental conditions and on a large scale is one of the strategies proposed to face this reality, either to generate such species that are in danger of extinction or to obtain potentially therapeutic compounds.

Achieving a comprehensive knowledge about the healing potential of *S. angustifolia* also contributes to the ecology field since it provides alternatives for the protection of the species and, therefore, to the conservation of biodiversity and cultural diversity of the country. This issue is a priority today. This review article highlights the multidisciplinary studies that the authors have undertaken and the achievements by adopting the research of *S. angustifolia*. This methodology traces ancestral knowledge toward modern biotechnology.

Biotechnological tools offer alternatives that are important to explore when it comes to obtaining medicinal plants in a massive way. Propagation through in vitro cultures that allow repopulating natural areas and conserving plant resources is an immediate option for the sustainable production of potential phytopharmaceuticals of interest. In the present work, the classic biotechnology tools were applied to establish the optimal conditions for micropropagation of *S. angustifolia*, procedures widely applied for the conservation of endangered medicinal species, a situation that does not exclude *S. angustifolia* [31]. In the same way, benefiting from the totipotentiality of the plant cell, the development of cell suspension cultures has enabled the generation of the scopoletin detected in the wild plant. Furthermore, it has led to the discovery of two new compounds of coumaric and naphthoic origin, whose anti-inflammatory activity is superior to the metabolites initially identified, and with an effect similar to that of naproxen and diclofenac, two non-steroidal anti-inflammatory drugs (NSAIDs) widely used to mitigate inflammation-related symptoms. Hairy root cultures of *S. angustifolia* were explored for stable and improved production of anti-inflammatory compounds [24].

## 5. Materials and Methods

This review article summarizes the experimental research conducted on the medicinal plant *Sphaeralcea angustifolia*, illustrating the substantial advances generated by its study. Published articles addressing *S. angustifolia* were identified using scientific search engines such as NCBI, ScienceDirect, Web of Science, and Google Scholar based on their titles and abstracts and then compiled and analyzed. The findings can be used to inform and optimize the strategies needed to apply the knowledge achieved and define the real therapeutic potential the plant offers, both for health care and the conservation and sustainable management of biodiversity.

## 6. Conclusions

Ethnopharmacological research on Mexican medicinal plants in Mexico explains, based on scientific support, the healing properties of the plants that the population uses to address health problems. Bibliographic research on the medicinal properties attributed to *S. angustifolia,* popularly known as vara de San José, confirmed its topical use to treat inflammatory processes and associated diseases. Scopoletin, identified in wild *S. angustifolia* plants, tomentin, and sphaeralcic acid de novo compounds produced from cells in suspension and hairy root in vitro cultures support the anti-osteoarthritic properties of a phytogel manufactured with the standard dichloromethane:methanol extract from the wild *S. angustifolia* plant. Toxicological evaluations in mice should be planned to evaluate the dichloromethane:methanol extract from the wild *S. angustifolia* plant. The micropropagation of *S. angustifolia* anticipates its conservation as its collection is controlled by the Mexican government. In the same way, cells in suspension and hairy root cultures were explored for the stable and enhanced production of anti-inflammatory compounds. Biotechnologically, industrial-scale processing of the cells in suspension has been achieved in a bioreactor stirring tank.

## 7. Patents

The studies carried out gave rise to two patents registered in Mexico (Instituto Mexicano de la Propiedad Intelectual, IMPI).

Romero-Cerecero, O., Tortoriello-García, J., Nicasio-Torres, M. del P.; Meckes-Fischer, M.E. del C.; Zamilpa-Álvarez, A. Composición farmacéutica que comprende el extracto de *Sphaeralcea angustifolia* para el tratamiento de osteoartitis 2019.

Nicasio-Torres, M.D.P., González-Cortazar, M., Meckes-Fischer, M., Tortoriello-García, J., Pérez-Hernández, J. Proceso biotecnológico de micropropagación y obtención de células en suspensión de *Sphaeralcea angustifolia* para la producción de nuevos compuestos con actividad antiinflamatoria 2019.

## Figures and Tables

**Figure 1 plants-12-00321-f001:**
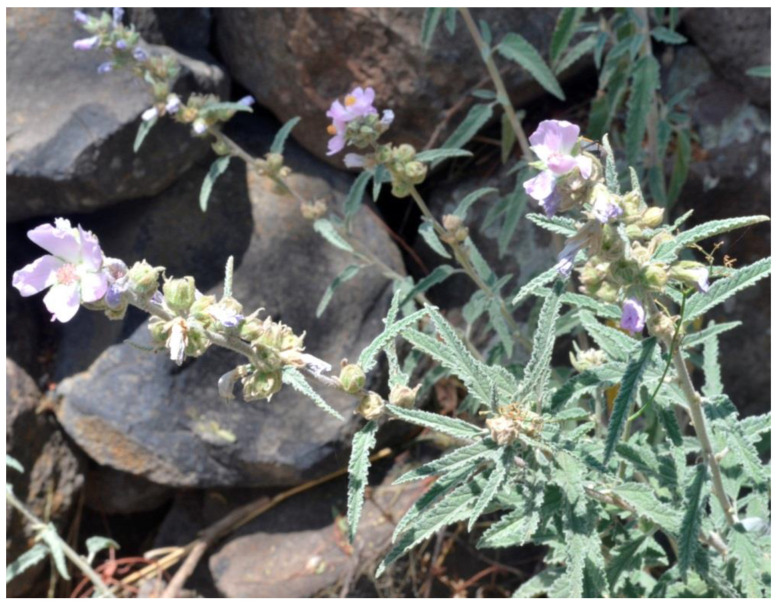
*Sphaeralcea angustifolia* wild plant growing in the municipality of Huichapan, Hidalgo State, Mexico.

**Figure 2 plants-12-00321-f002:**
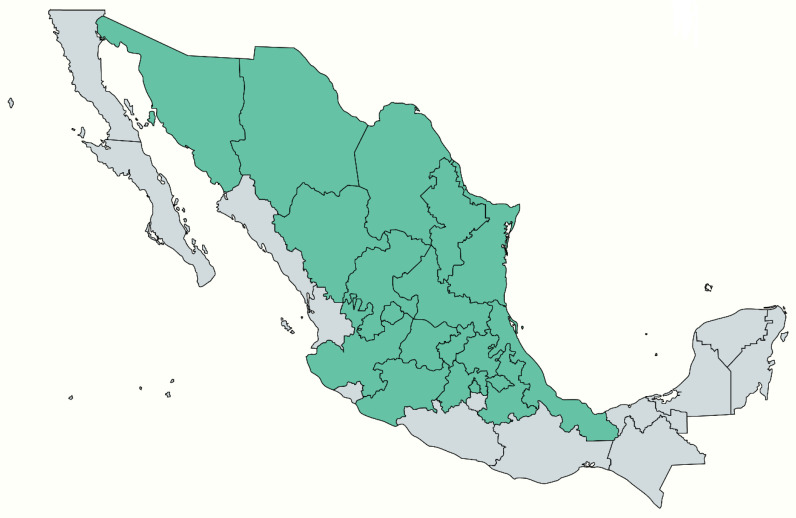
Geographical distribution of *Sphaeralcea angustifolia* species in Mexico.

**Figure 3 plants-12-00321-f003:**
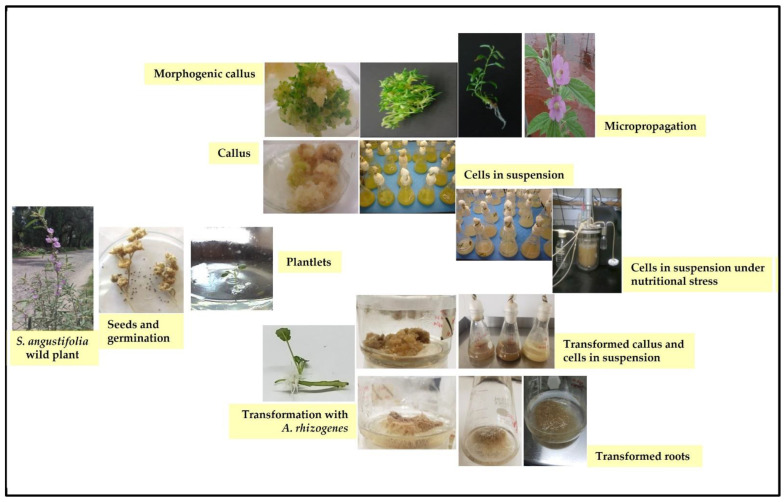
In vitro cultures established *Sphaeralcea angustifolia* as producers of active compounds.

**Table 2 plants-12-00321-t002:** Chemical structures of compounds identified in the *Sphaeralcea angustifolia* in vitro cultures.

Tissue	Extract	Biological Effect	Compounds		References
Cells in suspension	Dichloromethane:methanol	Anti-inflammatory Immunomodulatory	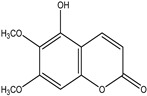	Tomentin	[16,19,20]
			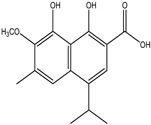	Sphaeralcic acid	[16,19,20]
			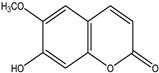	Scopoletin	[21,22,23]
Hairy roots	Dichloromethane:methanol	Anti-inflammatory Immunomodulatory	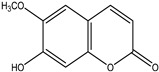	Scopoletin	[21,22,23]
			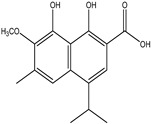	Sphaeralcic acid	[24]

## Data Availability

The data presented in the study are available in the article.

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
