# Peer review of "The Journey of a Medicinal Plant throughout Science: Sphaeralcea angustifolia (Cav.) G. Don (Malvaceae)"

_plants, 2023, doi:10.3390/plants12020321_

Round 1
Reviewer 1 Report
Dear Authors,
The written manuscript has scientific value. The obtained results for the species were adequately analyzed. A large number of studies included in the paper demonstrate a thorough analysis of the data. I suggest a few corrections before publishing the paper:
line 53 Sphaeralcea angustifolia, should be written in italics
line 138 cannot be said to have a different chemical composition, because it is understood that the type of extract is also different. Something can be said to be the same or different only if the same conditions apply. Change the context of the sentence.
line 234 explains the meaning of the abbreviation FDA
line 270 instead of Sphaeralcea angustifolia write S. angustifolia
I'm not completely clear about chapters 4, 5, and 6. Explain!
4. Material and methods. As the abstract says "This paper is a compendium based on the advances published in the scientific literature (from 2004 to 2021) on the anti-inflammatory properties of the plant and the potential offered by the species as a therapeutic agent, without dismissing aspects necessary to consider for the preservation of the resource and its cultural environment" should be written in detail in this chapter
5. the conclusion should be written
If you do not have a patent, delete this chapter
Author Response
Information is included in document attached.

Reviewer 2 Report
This review manuscript describes on the ethnobotanical background, chemical constituents, pharmacological evaluation, pharmacokinetic profiles, clinical trials, and biotechnological efforts for commercialization. Although it deals with original results, I’d like to reject this article for publication in the Plants because this paper has the following shortcomings.
i) First of all, it seems to be not adequate for a review paper because there are not sufficient previous data for this one species only. In addition, the phytochemical study on this plant needs the chemical structure, and it is preferable to summarize the data including chemical and pharmacological data into Tables or Figures to elevate the legibility. Overall the manuscript, it is impossible to find Figure 1 mentioned in Page 2.
ii) The authority of this plant should be checked again. It is preferable to add the information on the usage part of this plant in the description on the phytochemical and pharmacological data. The References Section should follow the “Instructions for Authors” of the Plants journal.
Author Response
Information is included in the attachement

Reviewer 3 Report
I have read the manuscript entitled "The course of a medicinal plant throughout science": Sphaeralcea angustifolia (Cav.) G. Don. (Malvaceae)," which summarizes the scientific work on Sphaeralcea angustifolia. Major revisions are needed to make the manuscript suitable for publication. In my opinion, the manuscript needs to be thoroughly revised so that it can be published
The specific comments and suggestions can be found in the manuscript file. The general comments are as follows:
1. I suggest changing the tittle to ‘The journey of a medicinal plant through science: Sphaeralcea angustifolia (Cav.) G. Don., Malvaceae"
2. The manuscript is very difficult to read and understand. Many sentences are too long and written in a too narrative manner.
3. The introduction section is too general. I suggest you give some information about the plant species you are discussing. Also, the introduction section does not cite the literature use.
4. In the conclusion, you should include a paragraph with a critical evaluation - something about the state of research on S. angustifolia, what gaps in knowledge were foun.
5. The manuscript contains grammatical and syntactical errors that need to be corrected.

Author Response
Information was included in the attachment

Reviewer 4 Report
NB - Detailed line by line comments are provided the attached PDF. The following is merely a summary of important points needing revision:
1. The introduction makes little mention of arthritis and no general summary of the condition has been made whereby the pathogenesis of arthritis is related to an inflammatory response. Anti-inflammatory activity appears to be the focus of the paper however specific mention is made of arthritis yet it is poorly discussed. A greater degree of clarity is required on what was the aim of this paper as it is poorly stated. If arthritis is the focus, then consider making the anti-inflammatory subject less broad.
2. Materials and methods pertaining to the acquisition of the scientific information presented in this review has not been adequately described.
3. Many sentences are excessively long and contain too many separate points. Consider breaking up these sentences for clarity
4. There is little mention of the controls used in some of the experimental trials mentioned. Controls allow for reproducibility and reliability of results. It is important to know what the plant extracts activity is being compared to and how is the 100% and 0% activity defined.
5. There is little mention on the use of statistical analyses used to confirm the statistical significance of the results. It is of great scientific value to determine whether results occurred by chance.
6. The topic of the review is relevant and its publication is likely to generate awareness on the therapeutic activity and need to conserve the species due its multi-factorial significance. The review highlights the importance of this medicinal plant species to a myriad of different role-players from traditional communities, scientists, cultural icons, traditional healers, pharmaceutical companies, conservation agencies and the global population alike.
7. Although the information presented is not necessarily new, it does combine a host of different sources aimed at bringing awareness to a focussed topic of anti-inflammatory activity. Having said this, the structure of the results section needs to be revised. There is a need for a greater sense of flow between adjacent sections.
8. There is a fairly well distributed mix of old and recent references.
9. A gap in knowledge has not necessarily been identified in this paper as most of this research has been presented before.
10. Ensure that all scientific names and phrases such as in vitro are in italics.
11. Consider including some images or figures and tables to increase the appeal of the review. It will also aid the reader in interpreting the results as a multitude of different experimental methodologies have been mentioned in the review.
12. The hypothesis of the study needs to be clearly stated, closely linked to the experimental methodology and furthermore, mention needs to be made on whether the hypothesis was confirmed or not. Poor mention of experimental design makes it difficult to determine whether the hypothesis may be accurately tested.
13. No excessive self-citation was detected.
14. The ethnobotanical usage of the plant seemed to be focused on stomach ailments. Elaborate further as to why anti-inflammatory activity was initially tested and what the justification was for focusing on a certain biological activity that was not commonly related to its traditional usage.

Author Response
Information is in the attachment

Round 2
Reviewer 2 Report
Where can I find the Figures in the revised manuscript? Nevertheless, the authority of this plant still has an error. Sphaeralcea angustifolia (Cav.) G.Don is the correct botanical name with authority. This paper seems not to describe sufficient previous data with this one species only for a review paper.
Author Response
It was written Sphaeeralcea angustifolia (Cav.) G.Don.
It was changed by Sphaeralcea angustifolia (Cav.) G. Don
Attachment with tables y figures was send to him

Reviewer 3 Report
The manuscript initially entitled 'The course of a medicinal plant throughout science: Sphaeralcea angustifolia (Cav.) G. Don. (Malvaceae)' has been have corrected more or less according to the suggestions, therefore, the manuscript can be accepted for publication.
Author Response
There are not comments. The reviewer: the manuscript can be accepted for publication.